# Psychosocial Risks and Violence Against Teachers. Is It Possible to Promote Well-Being at Work?

**DOI:** 10.3390/ijerph16224439

**Published:** 2019-11-12

**Authors:** Sabrina Berlanda, Marta Fraizzoli, Federica de Cordova, Monica Pedrazza

**Affiliations:** Department of Human Sciences, University of Verona, via San Francesco 22, 37129 Verona, Italy; marta.fraizzoli@univr.it (M.F.); federica.decordova@univr.it (F.d.C.); monica.pedrazza@univr.it (M.P.)

**Keywords:** violence against teachers, teachers’ well-being, social support, job demand, job control

## Abstract

Teaching has been reported to be one of the most stressful occupations, with heavy psychological demands, including the need to develop positive relationships with students and their parents; relationships that, in turn, play a significant role in teachers’ well-being. It follows that the impact of any violence perpetrated by a student or parent against a teacher is particularly significant and represents a major occupational health concern. The present study examines for the first time the influence of the Job Demands-Control-Support Model on violence directed against teachers. Six hundred and eighty-six teachers working in elementary and high schools in north-east Italy completed an online, self-report questionnaire. Our findings reveal the role played by working conditions in determining teachers’ experience of violence: greater job demands are associated with most offense types, whereas the availability of diffused social support at school is associated with lower rates of harassment. Workload should be equally distributed and kept under control, and violence should gain its place in the shared daily monitoring of practices and experiences at school in order to provide a socially supportive work environment for all teachers.

## 1. Introduction

The role of the teacher is viewed as somehow extending beyond the basic performance of their job [1]. Teachers are expected both to be role models and to somehow protect and take responsibility for students in a relationship of care [2,3]. Teachers are usually motivated by the desire to work with, and for the good of, other people, see students learn and develop, and make a difference in their lives. Indeed, the reasons given for choosing to train as a teacher can typically be classified as values, ethical motives, and intrinsic motivation [4]. However, although teachers typically view their role ultimately as being pastoral and instructional, it also involves maintaining discipline and dealing with student disruption [5].

Promoting well-being among teachers is an ethical concern [6], but it is also in the interest of students and society as a whole, since it affects the quality of education they provide [7,8]. Well-being is a multidimensional construct [9] that is subject to a range of personal, environmental, and relational factors [10,11]. Occupational well-being among teachers has not only been defined in terms of the presence of positive aspects [9], but also as the absence of negative factors such as, for instance, the absence of stress, which can arise from various sources such as numerous and onerous demands, a low level of autonomy, a lack of “social support” [3], and difficult relationships with students and their parents [12].

Multiple studies [13,14,15,16] have identified teaching as being one of the most stressful occupations, one with the potential to cause poor health. Indeed, compared to other professions, teachers have been found to achieve some of the lowest scores for physical health and psychological well-being [17]. Educational contexts are emotionally demanding [18,19,20]. Teachers are exposed to unrealistic expectations on the part of society, especially from parents [13,21], and to a substantial level of responsibility [13]. Furthermore, teachers are frequently faced with challenging situations and a lack of adequate resources, both material and immaterial [22,23]. Teachers spend a large amount of time preparing for lessons, grading tests, completing their administrative duties [13], and participating in frequent meetings with colleagues [17] and parents [15]. Other challenges include teaching unmotivated students and maintaining discipline in the classroom [16]. A new problem that teachers are increasingly having to contend with is violence perpetrated by students and their parents. In recent years, there has been a growing interest around the world in the impact of such abuse against teachers [1,5,24].

### 1.1. Job Demands-Control-Support Model

When dealing with risk factors, such as the demands placed on a professional or a lack of social support, which are neither tangible nor explicitly defined––being, rather, features of the psychosocial work environment––evaluating their adverse consequences in terms of well-being is complicated [25]. Since it was developed, the job demands-control-support (JDCS) model [26,27] has been widely used to explain the relationships between the psychosocial aspects of work and work-related well-being and health [6,25,28,29,30,31]. In 1979, Robert Karasek introduced the job demand-control (JDC) model, which comprises two basic dimensions: psychological job demands and control [25,32].

Job demands refer to task requirements in the work setting (or workload) and have been operationalized in terms of the amount of work, time pressure, unexpected tasks, and conflicting and emotional demands [25,30,31,33]. Job control is defined as the worker’s opportunity to organize their work and adopt their own initiatives. It also encompasses the opportunity to learn new things and develop skills [6,25,30,33]. According to Karasek and Theorell [27], psychological strain results not from single aspects of the work environment, but from a combination of onerous job demands and low job control that lead to a reduction in well-being and induce strain [25,30], which is considered a form of chronic stress [14].

Ten years later, Johnson and Hall [26] expanded the model with a third predictor of well-being and strain: workplace social support [25,32]. Social support at work can be defined as an assistive function performed by co-workers and supervisors in the form of meaningful practical, emotional, informational, and instrumental support in stressful situations [31,33,34,35]. This extended model became known as the job demand-control-support model (JDCS) [26,27]. Two hypotheses have been derived on the basis of this model: the (iso)strain hypothesis, which states that a combination of heavy job demands, limited job control, and a lack of social support are stressful and negatively predictive of workers’ well-being [6,25,30,31,33,36], and the buffer hypothesis, which predicts that control (and social support) can provide a buffer against the potential negative effects of heavy demands on well-being and health [25,33,36].

A European study involving high school teachers from 13 countries was carried out based on the JDCS model [37]. The results indicate that excessive job demands represent an important psychosocial risk that undermines teachers’ well-being [25]. Teaching places a large number of cognitive, emotional, and physical demands on the professional [31], including heavy workload, time pressure, interacting with students, parents and colleagues, and classroom management tasks. Other demands include administrative duties [13,15,17], frequent meetings, frequent communication with parents, and involvement in a large number of school development projects [15]. Job control is dependent on the administrative policies and leadership styles that characterize the schools in question. Usually teachers have a large degree of decision latitude and authority in their workplace relationships, as well as considerable job autonomy and the opportunity to develop work-related skills [31]. Social support from colleagues and supervisors is helpful in developing a working community where problems can be discussed and shared, and potential solutions considered collaboratively [38]. The effects of job autonomy, job demands, and social support on the physical and mental health of teachers has been explored in the literature [14,30,31]. Research has shown that both job autonomy and social support contribute to teachers’ satisfaction levels and well-being and are negatively related to burnout. Furthermore, high levels of social support lead to improvements in performance, while low levels are associated with negative outcomes in terms of teacher well-being [30].

### 1.2. Violence Toward Teachers

The student-teacher relationship plays a significant role in the level of well-being experienced by teachers [1,39]. It follows that the impact of any violence perpetrated by a student against a teacher will be particularly significant [40]. Across the globe, teacher-directed violence is an acute and serious issue that has received relatively little attention in the literature [41,42,43,44], with few studies exploring the violent mistreatment of teachers by students and students’ parents [2]. In a study conducted by the American Psychological Association Task Force exploring Classroom Violence Directed against Teachers, 80% of teachers across 48 states reported experiencing at least one instance of at least one type of victimization in the workplace over the course of the most recent school year [41,44,45,46]. A meta-analysis that included 24 studies shows that the prevalence of any type of teacher-reported violent victimization over the last two years ranged from 20% to 75% with a pooled prevalence of 53% [42]. The incidence of this sort of mistreatment of teachers is of particular concern since the literature suggests that, for a number of reasons, violence against teachers is actually under-reported [47]. The literature also highlights a growth in numbers of aggressive acts directed at teachers across different school types and locations [48,49], with students and students’ parents being the most common perpetrators [2,7,50].

Violence directed against teachers, perpetrated by students and/or their parents, refers to a range of aggressive behavior [41,44] that include insults and mockery [51], inappropriate comments and deliberate insolence [7], shouting and yelling [39], intimidation and verbal threats, harassment through the internet, damage to or theft of personal property, and physical assault [52]. Available studies agree that verbal violence was reported more often than other types of violence [46,53,54].

The research indicates that violence directed against teachers is a global phenomenon with similar features and outcomes present around the world [24]. These experiences have a negative impact on the general well-being, affecting physical, mental, and emotional health [2,41]. Such subjective conditions can also interfere with the professional role, leading teachers to develop negative attitudes toward school. Teachers who experience violence could develop a negative attitude toward their profession, such as discouraging the development of their professional abilities, or reducing their motivation [50] and commitment [49]. Furthermore, violence directed against teachers does not only affect the direct victims, but also those who witness it [50]. It also has an impact on the learning environment, standards of teaching, and, ultimately, the quality of education provided [2,41,49,50], and has severe negative consequences for the well-being and performance of students [2]. There are a number of additional costs in the form of absenteeism, lost instructional time, costs arising from medical and psychological care, and those incurred in training and the replacement of teachers who leave the profession [50].

Given the prevalence and negative impact of psychological strain and the experience of violence among teachers, research is needed to better understand teachers’ experience with these phenomena. The study of teacher-directed violence is still in its infancy [42,46] and we are not aware of studies that have explored the relationship between psychosocial risks factors (Demand–Control–Social support model) and violence perpetrated by students and their parents in the school context. So, our starting research questions were the follow: Do teachers perceive violent behavior perpetrated by pupils and/or their parents? How often? And, if they do, are level of job demands, job control, and social support factors associated with their perception of violence?

## 2. Materials and Methods

This study was developed according to the Code of Ethics issued by the Italian Psychological Association (AIP), and it has been approved by the Ethics Committee at the Department of Human Sciences at the University of Verona. All the procedures have been applied to guarantee privacy and anonymity of the participants according to Italian law.

### 2.1. Participants and Procedure

A group of 1360 teachers belonging to 14 primary and secondary schools was contacted by email, after their principles provided the researchers with their email addresses. After explaining the topic of the investigation, regarding psychosocial working conditions and the perception of violence toward teachers, they were asked to fill out a self-report online questionnaire. Between April and June 2019, 686 subjects completed the questionnaires on a voluntary basis (50.44% rate of reply). The 686 participants included 526 females (76.7%) and 155 males (22.6%), while 5 participants did not indicate a gender (0.7%). The mean age of the sample was 45.79 years (SD = 9.60; range = 25–67; 16 missing data, 2.33%). The mean of the number of years in the teaching profession was 16.14 (SD = 11.05; range = 1–43; 10 missing data, 1.46%). Most of the participants were working in senior-high schools (304, 44.31%), 200 in elementary schools (29.16%), and 179 participants were teaching at the junior-high school level (26.09%); 3 participants did not indicate a type of school (0.44%).

### 2.2. Measures

The questionnaire included questions on demographic and occupational characteristics (age, length of service, and type of school), types of violence in the workplace perpetrated by students and their parents, and psychosocial working conditions.

*Violence Toward Teachers.* The main tool adopted to record the perception of teacher-directed violence was the scale developed by McMahon, et al. [45], assessing the following forms of violence: (1) Harassment (5 items; e.g., “verbal threats”), (2) Property offenses (2 items; e.g., “damage to personal property”), (3) Physical attacks (4 items; e.g., “objects thrown”). Furthermore, participants were also asked to rate the frequency they had experienced or witnessed any form of violence directed at a teacher in the 12 months prior to completing the questionnaire. Their answers were both in relation to violent acts perpetrated by students, and in relation to violent acts perpetrated by students’ parents. Responses were given on a 4-point Likert scale that ranged from 1 (never) to 4 (frequently). The Cronbach’s alpha coefficients ranged from 0.621 to 0.841. The internal consistency values were 0.806 (harassment perpetrated by a student), 0.810 (property offenses perpetrated by a student), 0.621 (physical attacks perpetrated by a student), 0.803 (harassment perpetrated by a parent), 0.841 (property offenses perpetrated by a parent), and 0.778 (physical attacks perpetrated by a parent).

*Psychosocial working conditions*. The instrument used for the evaluation of psychosocial conditions at work according to the demand-control-support model was the Demand Control Support Questionnaire (DCSQ) [34,55,56] in the Italian versions [57]. The DCSQ is a reduced version of the Job Content Questionnaire (JCQ), covering 15 items distributed across three dimensions: psychological demands (five items; e.g., “My job requires working very hard”), control over the work process (four items; e.g., “I have an opportunity to develop my own special ability”) and social support at work (six items; e.g., “My co-workers are helpful in getting the job done”). Participants’ responses were recorded on a 4-point Likert scale, ranging from 1 (completely disagree) to 4 (completely agree). The Cronbach’s alpha coefficients ranged from 0.635 to 0.745: internal consistency values were 0.635 for *psychological demand*, 0.724 for *control over the work process*, and 0.745 for *social support*.

### 2.3. Data Analyses

Using SPSS 25 (IBM, Armonk, NY, USA), we performed descriptive statistics and reliability analysis (Cronbach’s alpha) for each scale. We computed a score for each variable by averaging the respective items. Pearson correlation was used to examine the relationship between variables. Furthermore, independent t-test was employed to evaluate whether different levels of teacher-directed violence, psychological demands, control over the work process, and social support at work were reported according to the variable “gender” and “length of service”. In addition, One-way ANOVA with post hoc Tukey was applied in order to search more in-depth potential differences according to the school category. Multiple linear regression analyses were applied to test the relationship between psychological demands, control over the work process, social support at work, length of service, gender, and violence perpetrated (harassment, property and physical offenses) by students or students’ parents.

## 3. Results

### 3.1. Descriptive Statistics and Correlations

Our results indicated that 84.8% of the teachers have experienced school-related violence in the 12 months prior to completing the questionnaire. *Harassment* was the most frequent type of student-and parent-perpetrated violence against teachers reported (80.6% of teachers had experienced this type of violence), followed by *physical attack* (48.5% of teachers had experienced this type of violence) and *property offense* (35.9% of teachers had experienced this type of violence). The most often reported offense type is *intimidation by students* (*M* = 2.11, *SD* = 0.95), followed by *obscene remarks by students* (*M* = 1.96, *SD* = 0.75) and *objects thrown by students* (*M* = 1.59, *SD* = 0.77). The results indicated that offenses perpetrated by students’ parents were less common (Table 1).

The means, standard deviations, and correlations of study variables are presented in Table 2. The data analysis shows that psychological demands are positively correlated with higher perceived levels of student- and parent-perpetrated violence. Regarding social support and job control, the results reveal that student-and-parent perpetrated violence is negatively correlated with perceived levels of job control and social support. Moreover, the perceived level of social support decreases as the length of service increases, and there is a positive correlation between teachers’ length of service and the weight of job demands.

### 3.2. Independent T-Test and One-Way ANOVA

When considering the variable “gender” (Table 3), we recorded that male teachers experience more social support (*M* = 2.86, *SD* = 0.58; *p* < 0.020) than female colleagues (*M* = 2.73, *SD* = 0.57), who perceive higher levels of parent-perpetrated harassment (*M* = 1.22, *SD* = 0.34) than male teachers (*M* = 1.13, *SD* = 0.26; *p* < 0.001). Female professionals also report a more onerous level of job demands (*M* = 2.69, *SD* = 0.58) than male teachers (*M* = 2.49, *SD* = 0.62; *p* < 0.001). Similarly, breaking down the results by length of service (Table 3), we can observe that new teachers feel stronger social support (*M* = 2.83, *SD* = 0.56) from colleagues and supervisors than longer-serving teachers (*M* = 2.70, *SD* = 0.59; *p* < 0.005). Nonetheless, less long-serving teachers report to feel more exposed to the risk of student-perpetrated harassment (*M* = 1.75, *SD* = 0.60; *p* < 0.010), while longer-serving professionals perceive a lower risk of student-perpetrated harassment (*M* = 1.63, *SD* = 0.53). Moreover, longer-serving teachers reported a more demanding level of job demands (*M* = 2.72, *SD* = 0.59) and lower levels of job control (*M* = 3.47, *SD* = 0.47) than new teachers (*M* = 2.58, *SD* = 0.60; *p* < 0.005; *M* = 3.55, *SD* = 0.40; *p* < 0.050). Breaking down the results by school level revealed additional significant differences (Table 3). Teachers at the elementary school level perceived a greater frequency of student-perpetrated physical attacks (*M* = 1.34, *SD* = 0.40) and parent-perpetrated harassment (*M* = 1.29, *SD* = 0.38) than teachers in both junior-high (*M* = 1.24, *SD* = 0.35; *p* < 0.020; *M* = 1.20, *SD* = 0.30; *p* < 0.050) and senior-high school (*M* = 1.19, *SD* = 0.31; *p* < 0.001; *M* = 1.16, *SD* = 0.30; *p* < 0.001). Furthermore, elementary school teachers perceived a greater burden of job demands (*M* = 2.79, *SD* = 0.55) than junior-high school teachers (*M* = 2.65, *SD* = 0.57; *p* < 0.050) and senior-high school teachers (*M* = 2.56, *SD* = 0.62; *p* < 0.001). Finally, senior-high teachers perceived lower levels of job control (*M* = 3.46, *SD* = 0.47) than junior-high teachers (*M* = 3.57, *SD* = 0.40; *p* < 0.050).

### 3.3. Multiple Regression Analysis of Variables on Violence

The results of multiple regression analysis (Table 4) show that student-perpetrated violence against teachers is negatively associated with perceived levels of social support from colleagues and supervisors, and is positively associated with greater job demands. Moreover, there is a negative association between the teacher’s length of service and both harassment and physical attacks by students. With regard to violence perpetrated by parents against teachers, there is a negative association between job control and both physical attacks and property offenses: higher levels of work control were associated with lower levels of violence. Harassment by students’ parents’, meanwhile, is positively associated with both greater job demands and being female, while being negatively associated with social support.

## 4. Discussion

The results of our study confirm that violence directed against teachers is very common. Indeed, it has been experienced by most of the teachers in the sample. According to McMahon, et al. [45], harassment by students is the most frequent form of violence against teachers. The data indicate that intervention programs need to focus on promoting the quality of teacher-student interactions as a major prevention variable [2]. Our data show that a high rate of professionals (84.8%) have declared to have been involved directly or indirectly in violent behavior, consistently with the international literature on the subject [45]. However, looking more in depth at the mean values per forms of victimization, the occurrences tend to be rather low. Namely, although the majority of teachers perceive violence at some level, its frequency is rare and mostly experienced in a mild form. Therefore, our findings suggest that uncommon and not necessarily severe acts of violence impact nonetheless heavily on teachers. Such a discrepancy offers some insight, and a hypothesis on possible explanations is worth noting. First, school is a peculiar social context where relationships between teachers and pupils, as well as between teachers and parents, are strictly constrained by behavioral scripts and the educational aims permeating this environment. The case of the Italian school model, specifically, is participatory and focused on the school-family alliance. Within this frame, any aggressive action and discourse oriented to teachers can be perceived as a relevant transgression of the social norm. In so doing, the behavior, per se, particularly when enacted by parents, can be considered a serious violation of the school-family alliance, regardless of its concrete consequence. Then, the perception of what is understood as a subversive act can leave teachers bewildered. In this sense, we believe that our findings reveal a raw nerve regarding a crisis of the current Italian school model and the teachers’ role in particular. For this reason, we can imagine that in the face of violence teachers feel unprepared and lacking the tools to deal with it. As things stand, ignoring violence until it concretely shows up in the professional’s closer circle can be considered a coping strategy, not totally effective, and making them vulnerable. This situation possibly mirrors a lack of conceptualization of violence at school, especially of the complexity of causes involved.

Breaking down the results by gender we see that, in line with previous studies [46], female teachers were more likely to report a greater number of experiences of parent-perpetrated harassment (experienced directly or witnessed) compared to male teachers. This would appear to be in line with the conclusion that, in comparison to men, women tend to perceive themselves as more vulnerable when exposed to threatening contexts [58,59]. In contrast to Chris Verhoeven, et al. [37], but consistent with Marcela Maria Birolim, et al. [34], female teachers are more vulnerable to greater job demands and low levels of social support than male teachers.

In line with several other studies [46,59], according to our data, less long-serving teachers were more inclined to feel at risk in their relations with students. This result is not easy to interpret because there are multiple variables that have not been considered in this study. The longer-serving teachers’ perception of a lower frequency of violent behavior might be explained, at least in part, by a greater ability to deal with student conflicts and manage their classrooms [46,59]. Also, more experienced teachers may elicit more respect from their students [59]. With regard to the Job Demands-Control-Support Model, it is interesting that, in our analysis, it emerges that more experienced teachers perceive a greater level of job demands and lower levels of control and social support from colleagues and supervisors. The negative association between length of service and social support is in contrast with previous research [60]. One possible explanation of this result is that longer-serving teachers tend to be more capable at managing problems, leading their colleagues to offer less support. Likewise, the positive association between length of service and job demands also diverges from the findings of previous research [37]. In this case, we may hypothesize that there is a positive relationship between the amount of experience a teacher has and the level of responsibility placed on him or her in the school setting.

In line with Gerberich, et al. [61], our data supports the finding that working with elementary school classes increases the risk of violence. Specifically, the elementary teachers in our study reported the highest levels of physical attacks by students and harassment by students’ parents.

Despite initiatives implemented to address the serious issue of violence in schools, it remains a matter of increasing concern. More work is needed to understand patterns of efficacy and distress over time in reaction to student misbehavior directed at teachers [59]. Our study explores for the first time the influence of the Job Demands-Control-Support Model on violence directed against teachers. The results suggest that in order to improve well-being at work and decrease perceived violence, it is necessary to promote social support at school and levels of job control among teachers. With regard to the first component of the model, greater job demands were associated with most offense types, indicating that the weight of job demands is of central importance for the well-being of teachers. Regarding the two other components, job control was only associated with property offenses and physical attacks perpetrated by students’ parents, while––in keeping with Gregory, et al. [62]—social support was associated with lower rates of harassment perpetrated by students or their parents. Receiving positive feedback from colleagues and supervisors, and the possibility of discussing experiences of violence are helpful in promoting well-being [13,33,63,64]. When colleagues and principals provide support, including emotional support, teachers report less harassment, which should be remembered is the most common type of threatening experience [46,62]. In keeping with the Job Demands-Control-Support Model [32], one way for job design interventions to be effective in reducing violence against teachers and improving teachers’ well-being is by altering the balance between perceived demand, control, and support. Since it is clearly difficult to reduce the psychosocial demands of the job given the responsibility, work-overload, and other psychological burdens placed on teachers, it follows that the most profitable strategy is to increase levels of job control and social support as a way to directly counteract the effects of such demands as stressors and factors in the perceived experience of violence and mistreatment. Our findings reveal the role played by working conditions in determining teachers’ experience of violence: greater job demands are associated with most offenses types, whereas the availability of diffused social support at school is associated with lower rates of harassment. Workload should be equally distributed and kept under control, and violence should gain its place in the shared daily monitoring of practices and experiences at school in order to provide a social supporting work environment for all teachers. We deem that it is important to frame accurately and more in depth the violence-at-school phenomenon, which still remains under-investigated. More systematic research and cross-cultural investigations, as well as qualitative detailed studies are possible research paths to flank in order to gain a multifaceted comprehension of this phenomenon and support teachers to develop more effective strategies.

### Limitations and Future Studies

There were some limitations in this investigation. Given its exploratory nature, the schools involved were gathered on a convenience basis; furthermore, the teachers who answered the online questionnaire represent a self-selected sample, possibly biased by elements such as digital skills and personal sensitivity to the topic, etc. For all of these reasons, and because of the limited number of schools involved, we cannot consider such data generalizable, even if the results of this research are consistent with the most updated literature on the topic. For data acquisition, validated scales have been applied in order to reduce the bias connected to self-reported measurements. However, we prioritized the number of completed questionnaires instead of the quantity of contextual variables, limiting the number of items. We are aware that this choice potentially affected the capacity of this study to address the differences in the school contexts, which merit further investigation. Another limitation is the low internal consistency of some measures (such as physical attacks perpetrated by a student). Its low internal consistency is related to two items: “Physical attacks (Physicians visit)” and “Weapon Pulled”. These types of violence are not frequent in Italy; we used a measure developed in the United States, but future studies could benefit from the development of an Italian scale to asses violence toward teachers. Finally, this research is basically cross-sectorial, and a longitudinal study would provide more specific data at this point. Moreover, given that social support seems to play an important role in teachers’ perception of the levels of violence, especially in relation to harassment, it would be beneficial for it to be included in future investigations.

## 5. Conclusions

Although violence against teachers continues to occur at alarming rates, current knowledge is surprisingly limited on the key contextual conditions, risk factors, and protective factors at work [46]. Our study supports validity of the demand-control-support model in a population of Italian schoolteachers and its ability to predict violence directed against teachers. Our findings reveal the role played by working conditions––including aspects of the working environment, such as social support, working demands, and job control––in determining teachers’ experience of violence and harassment, and how these combine with demographic variables, such as length of service and gender.

## Figures and Tables

**Table 1 ijerph-16-04439-t001:** Descriptive statistics of the offense type perpetrated against teachers.

Variable	Perpetrated by Students			Perpetrated by Students’ Parents		
	n	M	*DS*	n	M	*DS*
Obscene Remarks	686	1.96	0.748	685	1.20	0.391
Obscene Gestures	683	1.47	0.710	683	1.04	0.226
Verbally Threatened	684	1.46	0.712	683	1.30	0.580
Intimidated	678	2.11	0.949	681	1.39	0.594
Internet victimization	679	1.15	0.446	681	1.09	0.331
Theft of Property	683	1.34	0.638	683	1.01	0.127
Damage to Personal Property	680	1.35	0.650	682	1.01	0.126
Objects Thrown	685	1.59	0.774	684	1.01	0.121
Physical attacks (no Physician Visit)	682	1.30	0.590	684	1.02	0.152
Physical attacks (Physician Visit)	684	1.07	0.294	684	1.01	0.094
Weapon Pulled	684	1.02	0.152	685	1.01	0.132

**Table 2 ijerph-16-04439-t002:** Mean, standard deviation, and correlation of variables.

Variables		M	SD	1	2	3	4	5	6	7	8	9
1	HS	1.69	0.56	−								
2	PrOS	1.35	0.59	0.462 ***	−							
3	PhOS	1.25	0.35	0.555 ***	0.417 ***	−						
4	HSP	1.20	0.33	0.362 ***	0.253 ***	0.319 ***	−					
5	PrOSP	1.01	0.12	0.114 **	0.116 **	0.244 ***	0.228 ***	−				
6	PhOSP	1.01	0.10	0.101 **	0.186 ***	0.223 ***	0.183 ***	0.658 ***	−			
7	JC	3.50	0.44	−0.076 *	−0.052	−0.059	−0.089 *	−0.157 ***	−0.144 ***	−		
8	JD	2.65	0.60	0.164 ***	0.164 ***	0.194 ***	0.214 ***	−0.003	0.022	−0.115 **	−	
9	SS	2.76	0.58	−0.166 ***	−0.183 ***	−0.142 ***	−0.191 ***	−0.026	−0.058	0.271 ***	−0.365 ***	−
10	LS	16.14	11.05	−0.083 *	0.005	−0.043	0.025	−0.049	−0.028	−0.064	0.116 **	−0.092 *

HS Harassment by Students; PrOS Property Offenses by students; PhOS Physical Offenses by students; HSP Harassment by students’ parents; PrOSP Property Offenses by students’ parents; PhSP Physical Offenses by students’ parents; JC Job Control; JD Job Demand; SS Social Support; LS Length of service.* *p* < 0.05, ** *p* < 0.01, *** *p* < 0.001.

**Table 3 ijerph-16-04439-t003:** Differences in the sample means.

Variables	Men	Women	*t*	Less Long-Serving	Longer-Serving	*t*	Elementary School	Junior-High School	Senior-High School	*F*
HS	1.71 (0.59)	1.68 (0.55)	−0.703	1.75 (0.60)	1.63 (0.53)	2.647 **	1.62 (0.50)	1.69 (0.57)	1.72 (0.60)	1.908
PrOS	1.34 (0.57)	1.35 (0.60)	0.184	1.32 (0.57)	1.37 (0.61)	−0.964	1.34 (0.56)	1.31 (0.57)	1.37 (0.62)	0.504
PhOS	1.20 (0.32)	1.26 (0.36)	1.771	1.27 (0.37)	1.23 (0.33)	1.496	1.34 (0.40)	1.24 (0.35)	1.19 (0.31)	11.991 ***
HSP	1.13 (0.26)	1.22 (0.34)	3.630 ***	1.19 (0.34)	1.22 (0.32)	−0.937	1.29 (0.38)	1.20 (0.30)	1.16 (0.30)	9.448 ***
PrOSP	1.02 (0.18)	1.01 (0.09)	−0.924	1.02 (0.16)	1.00 (0.05)	1.861	1.03 (0.19)	1.01 (0.05)	1.01 (0.07)	1.806
PhOSP	1.02 (0.17)	1.01 (0.07)	−0.859	1.01 (0.13)	1.00 (0.06)	0.703	1.03 (0.19)	1.00 (0.04)	1.01 (0.06)	2.224
JC	3.46 (0.49)	3.52 (0.42)	1.375	3.54 (0.40)	3.47 (0.46)	2.322 *	3.52 (0.42)	3.57 (0.40)	3.46 (0.47)	3.588 *
JD	2.49 (0.62)	2.69 (0.58)	3.711 ***	2.58 (0.60)	2.72 (0.59)	−3.155 **	2.79 (0.55)	2.65 (0.57)	2.56 (0.62)	8.965 ***
SS	2.86 (0.58)	2.73 (0.57)	−2.543 **	2.83 (0.56)	2.69 (0.59)	3.129 **	2.77 (0.55)	2.83 (0.55)	2.71 (0.61)	2.454

HS Harassment by Students; PrOS Property Offenses by students; PhOS Physical Offenses by students; HSP Harassment by students’ parents; PrOSP Property Offenses by students’ parents; PhSP Physical Offenses by students’ parents; JC Job Control; JD Job Demand; SS Social Support. * *p* < 0.05, ** *p* < 0.01, *** *p* < 0.001.

**Table 4 ijerph-16-04439-t004:** Multiple regression analysis.

Variables	Perpetrated by Students	Perpetrated by Students’ Parents
	*β* (*SE*)	*β* (*SE*)
Multiple regression analysis of variables on Harassment		
Job Control	−0.043 (0.050)	−0.040 (0.029)
Job Demand	0.120 ** (0.039)	0.084 *** (0.022)
Social Support	−0.124 ** (0.041)	−0.068 ** (0.024)
Gender	0.059 (0.052)	−0.073 * (0.030)
Length of service	−0.005 ** (0.002)	−0.001(001)
*R* ^2^	0.055	0.071
*F*	7.675 ***	10.222 ***
*df*	5670	5669
Multiple regression analysis of variables on Property Offenses		
Job Control	0.006 (0.054)	−0.044 *** (0.011)
Job Demand	0.120 ** (0.041)	−0.003(008)
Social Support	−0.151 *** (0.043)	0.002 (0.009)
Gender	0.034 (0.055)	0.010(011)
Length of service	−0.001 (0.002)	−0.001 (0.000)
*R* ^2^	0.047	0.030
*F*	6.563 ***	4.160 **
*df*	5669	5669
Multiple regression analysis of variables on Physical Offenses		
Job Control	−0.016 (0.032)	−0.031 ** (0.009)
Job Demand	0.098 *** (0.024)	0.001 (0.007)
Social Support	−0.046 (0.026)	−0.005 (0.007)
Gender	−0.037 (0.033)	0.010 (0.009)
Length of service	−0.003 * (0.001)	0.000 (0.000)
*R* ^2^	0.049	0.025
*F*	6.910 ***	3.393 **
*df*	5670	5670

β = unstandardized coefficient. * *p* < 0.05, ** *p* < 0.01, *** *p* < 0.001.

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
