# Peer review of "Psychosocial Risks and Violence Against Teachers. Is It Possible to Promote Well-Being at Work?"

_ijerph, 2019, doi:10.3390/ijerph16224439_

Round 1

Reviewer 1 Report

I find the paper  thorough and well written. The authors have done a respectful job!

It is written to a well-informed audience and therefore unnecessary in both line 10 and 44 to state about the 'on the Surface  teachers' work' and that 'it does not appear challenging'. To whom? People know, even uninformed, that Teaching is or can be challenging. Therefore rewrite. Line 56. Too strong expression to talk about the ‘mistreatment’ of teachers. Find other way to state it. I suggest abuse as a more appropriate term. Line 129. Too narrow that it should only affect teachers’ motivation to teach. Research find that violation of teachers influences their subjective feelings of well-being, not just limited to commitment and motivation regarding school and teaching. Broaden the perspective at this point. Line 141-144. Make this part into a more pointed research question. Line 267-269. The authors ought to stress one striking finding: In spite of their research finds the phenomenon ‘very common’, teachers do not themselves find it ‘a very common occurrence’. What can be possible reasons for this discrepancy? Make this into an interesting discussion. Does this make teachers more surprised when it actually happens and make them unprepared for this phenomenon? Are they ignoring it until it happens? The abstract must be made clearer. It ought to be more stringent including all the essential academic elements of the full-length paper, namely; the background purpose focus methods results and conclusions

In the last part, the authors should state the importance of their findings. What is the gap in the research world of teaching they are bringing forward? What do their findings mean for the practical world? Implications? Why is their findings important? What does this mean for the agents in the real world of teaching? What changes is necessary based on their interesting findings?

Make this clearer and they have an interesting and well-founded article your journal should be proud to present.

Author Response

I find the paper  thorough and well written. The authors have done a respectful job!

Point 1: It is written to a well-informed audience and therefore unnecessary in both line 10 and 44 to state about the 'on the Surface teachers' work' and that 'it does not appear challenging'. To whom? People know, even uninformed, that Teaching is or can be challenging. Therefore rewrite.

Response 1:  Line 10 and 44: In order to accept your suggestions, we delated both line 10 and 44.

Point 2: Line 56. Too strong expression to talk about the ‘mistreatment’ of teachers. Find other way to state it. I suggest abuse as a more appropriate term.

Response 2: Line 56 (in the new manuscript line 63): We have changed “mistreatment” and used “abuse” as you suggested.

Point 3: Line 129. Too narrow that it should only affect teachers’ motivation to teach. Research find that violation of teachers influences their subjective feelings of well-being, not just limited to commitment and motivation regarding school and teaching. Broaden the perspective at this point.

Response 3: Line 129 (in the new manuscript lines 133-141): Thanks for your suggestion we have deepened the influence of violation of teachers on well-being.

Point 4: Line 141-144. Make this part into a more pointed research question.

Response 4: Line 141-144 (in the new manuscript lines 152-158): We have changed our aims into a more pointed research question.

Point 5: Line 267-269. The authors ought to stress one striking finding: In spite of their research finds the phenomenon ‘very common’, teachers do not themselves find it ‘a very common occurrence’. What can be possible reasons for this discrepancy? Make this into an interesting discussion. Does this make teachers more surprised when it actually happens and make them unprepared for this phenomenon? Are they ignoring it until it happens?

Response 5: Line 267-269 (in the new manuscript lines 281-303): Thanks for highligthing this potentiality in our article. We have stressed our findings and argued some possible explanations regarding this data.  

Point 6: The abstract must be made clearer. It ought to be more stringent including all the essential academic elements of the full-length paper, namely; the background purpose focus methods results and conclusions.

Response 6: Thank for the insights given about the abstract. We have made clearer the abstract, according to your suggestions.

Point 7: In the last part, the authors should state the importance of their findings. What is the gap in the research world of teaching they are bringing forward? What do their findings mean for the practical world? Implications? Why is their findings important? What does this mean for the agents in the real world of teaching? What changes is necessary based on their interesting findings?

Response 7: In the last part (in the new manuscript lines 331-332 and 351-360): We have stated the importance of our findings and their implications for the practical world.

Make this clearer and they have an interesting and well-founded article your journal should be proud to present.

Response 8: Thank you very much for your review, that improves the overall quality of our paper.

Reviewer 2 Report

This article examines psychosocial risks and violence and their influence on teacher's well-being at work. I believe the article has a great potential to contribute to our knowledge and policy for schools and teachers. I believe the paper will improve if authors address the following issues:

(1) Include information on number of schools from which these teachers are recruited in section 2.1

(2) Internal consistency of some measures (e.g., physical attacks perpetrated by a student) appear to be low in section 2.2. This needs to be acknowledged as a limitation of this study in page 9 (section 'limitations and future studies). It would also be useful if authors put forward some suggestions on how this measures could be improved in future studies.

Author Response

This article examines psychosocial risks and violence and their influence on teacher's well-being at work. I believe the article has a great potential to contribute to our knowledge and policy for schools and teachers. I believe the paper will improve if authors address the following issues:

(1) Include information on number of schools from which these teachers are recruited in section 2.1

Response 1: Thank you very much for your suggestion. We have included information on number of schools in line 16.

(2) Internal consistency of some measures (e.g., physical attacks perpetrated by a student) appear to be low in section 2.2. This needs to be acknowledged as a limitation of this study in page 9 (section 'limitations and future studies). It would also be useful if authors put forward some suggestions on how this measures could be improved in future studies.

Response 2: Thanks for your suggestion we have added as a limitation of our study the low internal consistency of some measures, and we have suggested how to improve these measures in future studies (in the new manuscript lines 371-376).